# AGER-1 Long Non-Coding RNA Levels Correlate with the Expression of the Advanced Glycosylation End-Product Receptor, a Regulator of the Inflammatory Response in Visceral Adipose Tissue of Women with Obesity and Type 2 Diabetes Mellitus

**DOI:** 10.3390/ijms242417447

**Published:** 2023-12-13

**Authors:** Klaudia Gutowska, Krzysztof Koźniewski, Michał Wąsowski, Marta Izabela Jonas, Zbigniew Bartoszewicz, Wojciech Lisik, Maurycy Jonas, Artur Binda, Paweł Jaworski, Wiesław Tarnowski, Bartłomiej Noszczyk, Monika Puzianowska-Kuźnicka, Krzysztof Czajkowski, Alina Kuryłowicz

**Affiliations:** 1II Department of Obstetrics and Gynecology, Warsaw Medical University, 00-315 Warsaw, Poland; klaudia.gutowska@wum.edu.pl (K.G.); krzysztof.czajkowski@wum.edu.pl (K.C.); 2Department of Human Epigenetics, Mossakowski Medical Research Centre, Polish Academy of Sciences, 02-106 Warsaw, Poland; krzychukoz@gmail.com (K.K.); martajonas@imdik.pan.pl (M.I.J.); mpuzianowska@imdik.pan.pl (M.P.-K.); 3Department of General Medicine and Geriatric Cardiology, Medical Centre of Postgraduate Education, 00-401 Warsaw, Poland; mwasowski@cmkp.edu.pl; 4Department of Internal Medicine and Endocrinology, The Medical University of Warsaw, 02- 097 Warsaw, Poland; z.bartoszewicz@wum.edu.pl; 5Department of General and Transplantation Surgery, The Medical University of Warsaw, 02-005 Warsaw, Poland; wojciech.lisik@wum.edu.pl; 6Department of General Surgery, Barska Hospital, 02-315 Warsaw, Poland; morjon@poczta.onet.pl; 7Department of General, Oncological and Bariatric Surgery, Medical Centre of Postgraduate Education, 00-401 Warsaw, Poland; abinda@cmkp.edu.pl (A.B.); pjaworski@cmkp.edu.pl (P.J.); wtarnowski@cmkp.edu.pl (W.T.); 8Department of Plastic Surgery, Medical Centre of Postgraduate Education, 00-401 Warsaw, Poland; bnoszczyk@melilot.pl; 9Department of Geriatrics and Gerontology, Medical Centre of Postgraduate Education, 01-826 Warsaw, Poland

**Keywords:** advanced glycosylation end-products (AGEs), advanced glycosylation end-product receptor (AGER), long non-coding RNA (lncRNA), adipose tissue, obesity, metabolic inflammation, type 2 diabetes mellitus (T2DM)

## Abstract

The advanced glycosylation end-product receptor (AGER) is involved in the development of metabolic inflammation and related complications in type 2 diabetes mellitus (T2DM). Tissue expression of the AGER gene (*AGER*) is regulated by epigenetic mediators, including a long non-coding RNA AGER-1 (lncAGER-1). This study aimed to investigate whether human obesity and T2DM are associated with an altered expression of *AGER* and lncAGER-1 in adipose tissue and, if so, whether these changes affect the local inflammatory milieu. The expression of genes encoding AGER, selected adipokines, and lncAGER-1 was assessed using real-time PCR in visceral (VAT) and subcutaneous (SAT) adipose tissue. VAT and SAT samples were obtained from 62 obese (BMI > 40 kg/m^2^; *N* = 24 diabetic) and 20 normal weight (BMI = 20–24.9 kg/m^2^) women, while a further 15 SAT samples were obtained from patients who were 18 to 24 months post-bariatric surgery. Tissue concentrations of adipokines were measured at the protein level using an ELISA-based method. Obesity was associated with increased *AGER* mRNA levels in SAT compared to normal weight status (*p* = 0.04) and surgical weight loss led to their significant decrease compared to pre-surgery levels (*p* = 0.01). Stratification by diabetic status revealed that *AGER* mRNA levels in VAT were higher in diabetic compared to non-diabetic women (*p* = 0.018). Elevated *AGER* mRNA levels in VAT of obese diabetic patients correlated with lncAGER-1 (*p* = 0.04, r_s_ = 0.487) and with interleukin 1β (*p* = 0.008, r_s_ = 0.525) and resistin (*p* = 0.004, r_s_ = 0.6) mRNA concentrations. In conclusion, obesity in women is associated with increased expression of *AGER* in SAT, while T2DM is associated with increased *AGER* mRNA levels and pro-inflammatory adipokines in VAT.

## 1. Introduction

Adipose tissue is composed of heterogeneous anatomical depots with different epigenetic characteristics that determine their metabolic functions [1]. Although initially identified as a passive energy store, definitive studies have underlined its complex nature, ensuring endocrine, autocrine, and immune homeostasis [2]. Furthermore, adipose tissue is highly flexible and can therefore be remodeled according to caloric supply [1]. The excessive accumulation of energy reserves associated with obesity leads to adipocyte dysfunction, mitochondrial damage, and hypoxia, resulting in the disruption of cell metabolism and mechanical stress through stretching of the extracellular matrix [3,4]. It is widely accepted that obesity is associated with chronic low-grade inflammation, driven by various immune cells, which can result in irreversible organ damage [5]. In a state of positive energy balance, inflammatory cytokine and chemokine responses shift the immune system from an anti-inflammatory to a pro-inflammatory state and lead to an increased risk of developing metabolic disorders, including insulin resistance (IR) and type 2 diabetes mellitus (T2DM) [5]. Although this relationship is considered controversial by some, basic and clinical research confirms that the level of inflammation associated with obesity is positively correlated with the degree of IR and T2DM [6].

A key link between adipose tissue expansion, associated comorbidities, and inflammatory signaling is the advanced glycosylation end-product receptor (AGER) [7,8]. This cell surface molecule—present in a variety of cell types—triggers the generation of reactive oxygen species and the induction of an inflammatory response. Upon binding a variety of different ligands, including advanced glycation end-products (AGEs), it mediates the activation of nuclear factor κB (NF-κB) and other pro-inflammatory pathways, leading to tissue dysfunction and the development of obesity and other metabolic diseases [9,10,11].

Evidence suggests that activation of the inflammatory network depends on equally important environmental, genetic, and epigenetic contributing factors [12]. Long untranslated non-coding RNAs (lncRNAs) of more than 200 nucleotides are considered to be a hallmark of epigenetic regulation in inflammatory signaling [12]. These transcripts can act as regulatory elements that affect gene expression at different stages, including modulation of transcriptional availability of chromatin and mRNA processing [13,14]. Dynamic epigenetic modulations accompany obesity, as lncRNA expression patterns in adipose tissue and sera of obese subjects differ from those of normal weight individuals [15,16]. Furthermore, IR has been shown to alter non-coding RNA expression in obese subjects, confirming the central contribution of low-grade inflammation in both conditions [17]. As lncRNAs contribute to the epigenetic shaping of inflammatory genes, and AGER is a promoter of the immune response that is enhanced in obesity and T2DM, there may be a link between the concentration of particular lncRNAs and *AGER* expression.

With this in mind, in the current study, we aimed to investigate whether human obesity and T2DM are associated with altered expression of *AGER* and long non-coding RNA AGER-1 (lncAGER-1) in adipose tissue and, if so, whether these changes affect the local inflammatory milieu.

## 2. Results

### 2.1. Clinical and Biochemical Characteristics of Study Participants

The study group consisted of 62 women with a body mass index (BMI) of >40 kg/m^2^—calculated by weight in kg divided by height squared (m^2^). Based on the American Diabetes Association diagnostic criteria, T2DM was diagnosed in 23 participants (37.0%) within the study group [18]. All patients in the study group (O) underwent surgical treatment for obesity (sleeve gastrectomy or mini gastric bypass). During these surgical procedures, visceral (VAT) and subcutaneous (SAT) adipose tissue samples were collected from the lower abdomen. Fifteen additional SAT samples from the lower abdomen were obtained during abdominoplasty surgery, from previously obese (PO; BMI = 24.3–29.5 kg/m^2^) study participants who were 18 to 24 months post-bariatric surgery. Five patients in the post-bariatric group had residual hypertension, three had persistent pre-diabetes, and six had hyperlipidemia, but none met the criteria for a diagnosis of metabolic syndrome. As abdominoplasty is not associated with opening the abdominal cavity, it was not possible to obtain visceral adipose tissue samples from the PO participants.

The control group (N) consisted of 20 women with a BMI within the normal range (20.7–24.93 kg/m^2^). In addition, they had no history of chronic disease, and their metabolic health was confirmed as normal following a physical examination and biochemical blood tests. Although their body composition was not assessed, they were considered metabolically healthy based on their medical histories, normal BMI and biochemical parameters, and the absence of any metabolic syndrome components. None of the study participants were receiving anti-inflammatory treatment (e.g., glucocorticoids, non-steroidal anti-inflammatory drugs, or anti-cytokine drugs).

The baseline clinical and biochemical parameters of the study participants are summarized in Table 1.

### 2.2. The Advanced Glycosylation End-Product Receptor (AGER) Gene and Long Non-Coding RNA AGER-1 Expression in Adipose Tissue of Obese Individuals before and after Bariatric Surgery and Normal Weight Subjects

*AGER* expression at the mRNA level was higher in the subcutaneous adipose tissue of obese patients (SAT-O) compared to the SAT of normal weight individuals (SAT-N, *p* = 0.04). Surgically induced weight loss was associated with a significant decrease in *AGER* mRNA levels in SAT (SAT-PO) (*p* < 0.001). Interestingly, there were substantially higher *AGER* mRNA levels in SAT-O compared to the visceral adipose tissue in obese participants (VAT-O), whereas in normal weight women, no significant differences in *AGER* expression were found between SAT and VAT (Figure 1a). When the obese study participants were stratified according to diabetic status, we found that what distinguished women with T2DM from normoglycemic women was the higher expression of *AGER* at the mRNA level in VAT (*p* = 0.018, Figure 1b). 

The expression profile of lncAGER-1 in the studied tissues resembled that of *AGER* in many respects (Figure 1c). lncAGER-1 levels were significantly higher in the SAT of obese women compared to normal weight women (*p* < 0.001), and weight loss was associated with a marked decrease in lncAGER-1 levels (*p* = 0.004), but not to the level observed in the SAT of normal weight women (*p* = 0.005). There were differences in lncAGER-1 levels (Figure 1c) between adipose depots depending on the presence of obesity: in obese women, lncAGER-1 expression was higher in SAT (*p* = 0.02), whereas in normal weight women, it was higher in VAT (*p* = 0.003). Stratification of obese patients according to glycemic status revealed a similar trend in the expression profile of lncAGER-1 to *AGER* between the tissues examined, but the differences were not significant (Figure 1d).

Subsequent analysis (Figure 2) revealed a significant positive correlation between *AGER* mRNA and lncAGER-1 levels in the VAT of the obese diabetic subjects (*p* = 0.04, r_s_ = 0.487, Figure 2b). 

### 2.3. Expression of Pro-Inflammatory Adipokines in Adipose Tissue of Obese Subjects, Stratified by Diabetic Status

The finding that *AGER* mRNA levels differ between the tissues studied, and are higher in obese individuals (particularly those with T2DM), prompted us to investigate whether these differences affect the local inflammatory milieu. To this end, we examined the expression of genes that encode selected pro-inflammatory adipokines, namely interleukin 1 (*IL1B*), interleukin 6 (*IL6*), interleukin 8 (*IL8*), and resistin (*RETN*), at mRNA and protein levels, and correlated their concentrations with *AGER* mRNA levels. Due to the paucity of biological material, protein measurements were not performed in the PO group (18 to 24 months post-bariatric surgery). In our previous studies on a smaller group of patients, we found that the concentration of these adipokines in adipose tissue is significantly altered in the course of obesity [19,20]. Similarly, in the group of women studied in this work, we found that obesity was associated with an increase in the expression of genes encoding pro-inflammatory adipokines at both the mRNA and protein levels (Appendix A).

The study group was then stratified according to glycemic status, and adipokine mRNA levels in adipose tissue were compared between the diabetic and non-diabetic study participants (Figure 3).

This stratification revealed that the mRNA levels for *IL1B* (*p* = 0.04) and resistin (*p* = 0.02) were significantly higher in the VAT of the diabetic, obese study participants (VAT-D) compared to the non-diabetic participants (VAT-ND). However, glycemic status did not affect adipokine protein levels in a given adipose tissue depot (Appendix A). To verify whether the observed differences in adipokine mRNA levels between diabetic and non-diabetic subjects could be related to *AGER* expression, correlation analyses were performed. Significant positive correlations were found between *AGER* mRNA levels and *IL1B* (*p* = 0.008, r_s_ = 0.525) and *RETN* (*p* = 0.004, r_s_ = 0.60) mRNA levels in visceral adipose tissue of obese diabetic participants (Figure 4).

## 3. Discussion

Obesity-related adipose tissue dysfunction appears to play a critical role in the pathogenesis of metabolic complications, including insulin resistance and T2DM [1,4]. One manifestation of adipose tissue dysfunction is chronic inflammation, resulting from impaired adipocyte energy homeostasis that involves the extracellular matrix [4,5]. The key pathway in the development of metabolic inflammation and insulin resistance is the receptor for advanced glycation end-products’ (known as AGER or RAGE) signaling. Therefore, in this work, we investigated whether human obesity and T2DM are associated with altered expression of the AGER gene and its regulator lncAGER-1 in adipose tissue, and, if so, whether these changes affect the local inflammatory milieu. We found that obesity in women is associated with increased expression of *AGER* in subcutaneous adipose tissue, whereas AGER-mediated development of metabolic inflammation—which predisposes to increased insulin resistance and T2DM—preferentially affects the visceral depot.

Our finding regarding the elevated *AGER* expression in adipose tissue in obesity is consistent with the results of previous reports. Gaens et al. conducted a study on subcutaneous tissues—obtained from 10 patients with grade I/II obesity (mean BMI 34.2 ± 4.0 kg/m^2^) and from 9 normal weight controls—and found that *AGER* expression in both mRNA and protein levels is higher in the course of obesity [21]. Visceral adipose tissue was not analyzed in that study. However, when comparing tissues from different depots of obese patients, the authors found a significant increase in *AGER* mRNA levels in VAT compared to SAT. In our study, which focused only on women with obesity, we observed the opposite trend. As adipose tissue is a highly heterogeneous organ, its depots are biologically different, which is reflected in differences in the expression profiles of various genes, and translates into differences in lipolytic and inflammatory activity or insulin sensitivity [2]. In obesity, the functional differences between the different adipose tissue depots might be both accentuated and abolished [19,22], and these processes can be mediated by epigenetic mechanisms that buffer the effects of the environment [23]. There are many potential reasons for the discrepancy between our findings and the Gaens et al. study, such as differences in gender and metabolic status (in the study by Gaens et al., 30% of the participants were men and most of the patients had impaired glucose tolerance or T2DM) [21]. The latter aspect is important since, after stratifying the obese study participants according to glycemic status, we observed that VAT from diabetic subjects had higher levels of *AGER* mRNA than VAT of normoglycemic individuals. A similar association between T2DM and higher concentrations of *AGER* mRNA was previously observed in pericardial adipocytes [24]. 

Given the role of AGER in regulating the inflammatory response, we reasoned that the association between higher expression of this receptor in adipose tissue and T2DM might not be coincidental. Chronic low-grade inflammation is a primary event of obesity-related insulin resistance [25]. An essential role in inflammatory signaling in human adiposity is played by pro-inflammatory adipokines, including IL-1β, IL-6, IL-8, and resistin [19,20,26,27,28,29]. We demonstrated here, and before [19], that in the obese study participants, levels of mRNAs encoding these molecules were significantly increased in SAT. This reflects findings from other studies, wherein SAT, not VAT, was found to be a leader of cytokine synthesis [29,30,31]. The debate over which depot of adipose tissue plays the dominant role in the development of obesity-related inflammation has been ongoing for years, with several papers in the literature pointing to VAT as the main source of pro-inflammatory mediators [32,33,34]. Our observations suggest that patients’ glycemic status is crucial in this case. In our study, we observed higher *IL1B* and *RETN* mRNA in VAT of patients with obesity plus T2DM, rather than in obese patients with a euglycemic state. In preclinical studies, IL-1β has been identified as a key mediator of macrophage-induced insulin resistance in human adipocytes, and its high levels in VAT predispose obese rodents to the development of diabetes [35,36]. In addition, this cytokine orchestrates the mobilization of other inflammatory mediators, e.g., IL-6 contributing to insulin resistance and dyslipidemia [37]. In turn, resistin can be either the sender or recipient of pro-inflammatory processes by engaging the 5 AMP-activated protein kinase (AMPK) and AMPK-independent suppressor of cytokine signaling-3 (SOCS-3) pathways [38,39,40]. Subsequently, high levels of resistin correlate with insulin resistance, glucose intolerance, and endothelial dysfunction marker concentration [41]. In conclusion, when assessing the relationship between *AGER* expression and adipokine-encoding genes, it is crucial to define glycemic status, as different correlations can be expected in normoglycemic individuals and others with T2DM [42]. 

Our results suggest that AGER might be the chief of tri-directional communication between obesity, T2DM, and inflammation. In addition to inducing insulin resistance in adipose tissue, there is evidence that AGER ligands can induce pancreatic islet inflammation and beta cell apoptosis [43,44]. This raises the question of what mechanisms regulate *AGER* expression in obesity.

Previous studies proposed that lncRNAs, arising from noncoding transcripts, might contribute to the pathogenesis of metabolic disturbances, as well as cancer, neurodegenerative, and immune system diseases [45,46]. LncRNAs have a variety of functions; some (despite their name) encode peptides, but primarily, they exert post-translational effects on gene expression levels (both up-and down-regulated) and compete for the binding of specific miRNAs with mRNAs [47]. Here, we reported that lncAGER1, a functionally proven *AGER* regulator, may modulate its expression profile in adipose tissue. What’s more, obesity promoted an increase in lncAGER-1 level in SAT, whereas in normal weight subjects, higher expression of this transcript was found in VAT. Interestingly, tandem obesity and diabetes-induced increased levels of lncAGER-1 in VAT were positively correlated with *AGER* expression. Our findings indirectly suggest that lncAGER-1 may induce inflammation mediated by *AGER*, therefore predisposing individuals to increased insulin resistance and T2DM. Functional studies have proven that lncAGER-1 acts as a miRNA sponge and indirectly modulates the expression of *AGER* in lung or colorectal cancer [48,49]. However, this is the first report to suggest lncAGER-1 as a contributor to obesity-related insulin resistance and T2DM. In summary, our study reveals the possible role of lncRNA in the upregulation of *AGER* expression in the adipose tissue and therefore, indirectly, in the development of metabolic inflammation in the course of obesity; however, these phenomena seem to be depot-specific.

The main limitation of this study is its descriptive nature, which implies the need for further functional studies to confirm our findings. When describing the correlations between RNA and lncRNA levels, we can only surmise that they reflect tissue regulatory processes. Although the role of lncAGER-1 in regulation of *AGER* expression has been confirmed in functional studies in other tissues and organs, our results require in vitro verification in human adipocyte cell lines. Another limitation is the relatively small number of control tissues obtained from the normal weight individuals and patients who underwent bariatric surgery. On the other hand, the main strengths of the current study are its homogeneous, well-characterized population, with 62 participants in the study group, and its innovative nature, which can be considered a novel attempt to elucidate the mechanisms linking obesity and type 2 diabetes mellitus. 

## 4. Materials and Methods 

### 4.1. Tissue and Blood Sample Collection

Visceral (VAT) and subcutaneous (SAT) adipose tissue samples were collected from patients with obesity during bariatric surgery. SAT samples were obtained from the lower abdomen during abdominoplasty in previously obese study participants who were 18 to 24 months post-bariatric surgery. In the control group, VAT and SAT pairs were obtained at the time of elective cholecystectomy in a manner analogous to the study group. All adipose tissue samples were immediately frozen at −80 °C and then homogenized in liquid nitrogen. Before surgery, all participants underwent a medical examination and provided 15 mL of venous blood. All sera and adipose tissue samples were stored at −80 °C until molecular and biochemical measurements were performed.

This study was approved by the Bioethics Committee of the Medical University of Warsaw and the Centre for Postgraduate Medical Education in Warsaw (decision no. KB 147/2009 issued on 28 July 2009, KB 91/A/2010 issued on 19 July 2010, KB 117/A/2011 issued on 14 November 2011, and KB 38/A/2022 issued on 16 May 2022) and written informed consent was obtained from all participants. 

### 4.2. Isolation of Total RNA, Reverse Transcription, and Real-Time PCR

Total RNA was isolated and reverse-transcribed to cDNA using previously described methods [19]. Real-time PCR was performed in triplicate using a LightCycler 480 Instrument II (Roche, Mannheim, Germany), as previously described [50]. The specific primers used to analyze lncAGER-1 and gene expression at the mRNA level in adipose tissue are listed in Table 2.

### 4.3. Isolation of a Protein Fraction from Adipose Tissue and Measurement of Cytokine Concentrations

Isolation of the adipose tissue protein fraction was performed as previously described [19,20]. An ELISA-based chemiluminescent Q-plex Custom array (Quansys Bioscience, West Logan, UT, USA) was used to measure interleukin (IL) 1β, 6, 8, and resistin concentrations in adipose tissue protein extracts. Luminescence was assessed using a Molecular Imager Versa Doc™ MP 5000 system (Bio-Rad, Hercules, CA, USA), according to the manufacturer’s guidelines. Results were analyzed using Q-View software version 2.17 (Quansys Bioscience, West Logan, UT, USA). Measurements of interleukins and resistin concentrations in adipose tissue were normalized to total protein concentrations in protein extracts. Mean total protein concentrations in VAT and SAT extracts from obese and normal weight participants were not significantly different (*p* > 0.05).

### 4.4. Statistical Analysis

The normality of distribution and homogeneity of variance of the studied parameters were checked using the Shapiro–Wilk and Levene tests, respectively. Differences in mRNA and protein levels in the tissues studied were calculated using Student’s t/Mann–Whitney U tests. The Bonferroni adjustment for multiple comparisons was not applied [52,53]. The Spearman correlation test was used for correlations between quantitative values. All statistical analyses were performed using the Statistica software package v.10 (StatSoft, Tulsa, OK, USA) and GraphPad Prism software v.7 (GraphPad Software, San Diego, CA, USA).

## 5. Conclusions

In conclusion, our data suggest that obesity in women is associated with increased expression of *AGER* in SAT. However, AGER-mediated development of metabolic inflammation—which predisposes to increased insulin resistance and type 2 diabetes mellitus—preferentially affects VAT. Given the descriptive nature of our study, these data need to be verified in vitro.

## Figures and Tables

**Figure 1 ijms-24-17447-f001:**
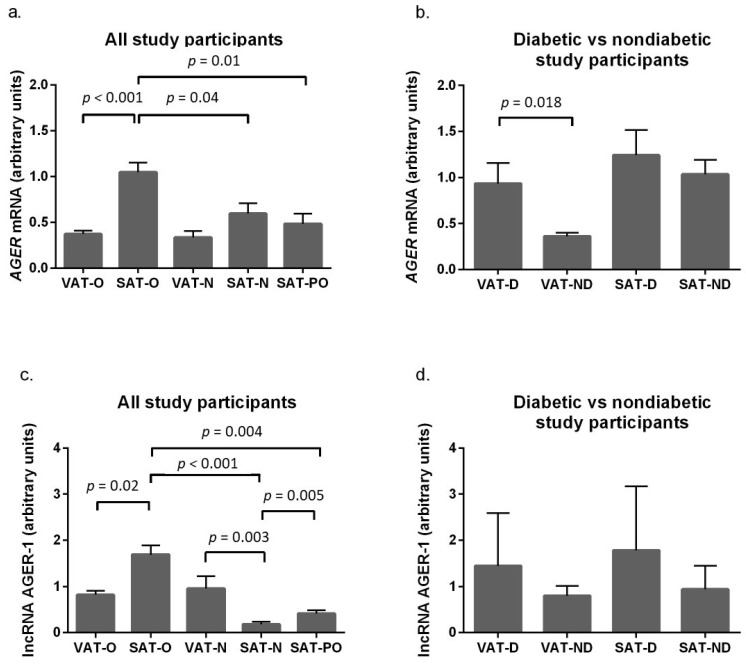
*AEGR* mRNA (**a**,**b**) and lncAGER-1 (**c**,**d**) levels in visceral (VAT) and subcutaneous (SAT) adipose tissue of obese subjects before (O) and after surgical weight loss (PO), in normal weight subjects (N) and diabetic (D) and non-diabetic (ND) obese study participants. Results are presented as median with interquartile range.

**Figure 2 ijms-24-17447-f002:**
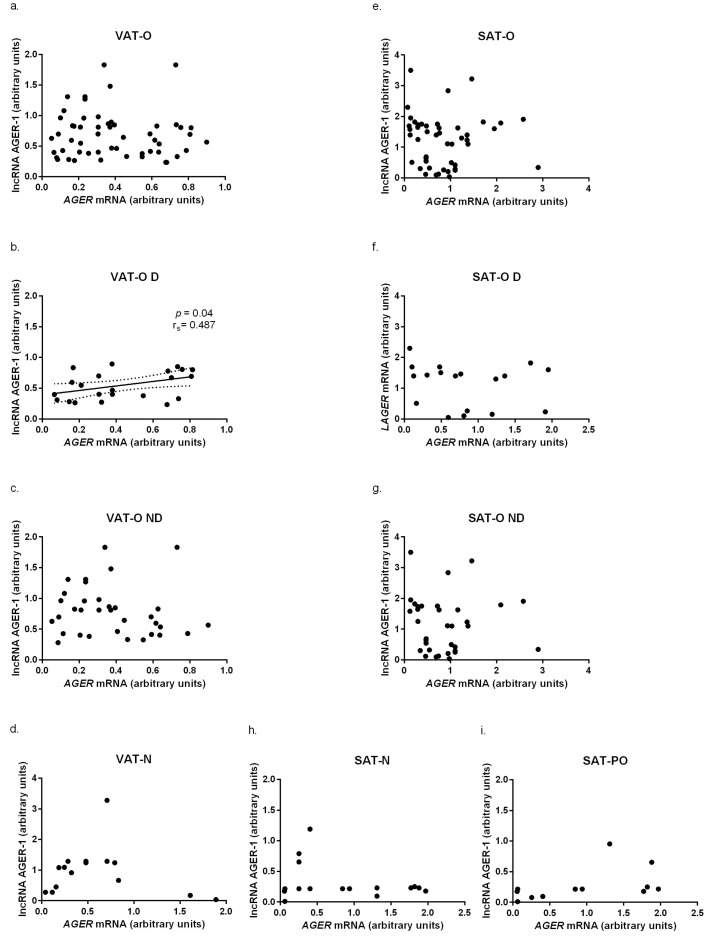
Correlations between AGER mRNA levels and AGER-1 lncRNA levels in visceral adipose tissue (VAT) and subcutaneous adipose tissue (SAT) of all obese (O), obese diabetic (D), obese non-diabetic (ND), normal weight (N), and post-bariatric surgery (PO) study participants. Black dots represent particular participants. Lines represent linear regression analysis with 95% CI interval. (**a**–**i**) refer to the phenomena observed in the different fat deposits in the different study groups.

**Figure 3 ijms-24-17447-f003:**
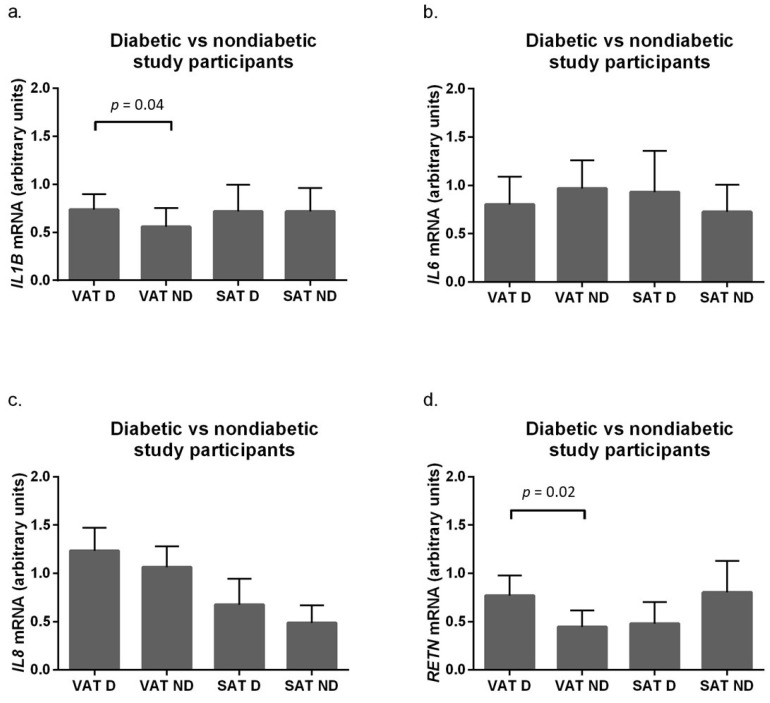
mRNA levels of genes encoding (**a**) interleukin 1b (*IL1B*), (**b**) interleukin 6 (*IL6*), (**c**) interleukin 8 (*IL8*), and (**d**) resistin (*RETN*) in visceral (VAT) and subcutaneous (SAT) adipose tissue of diabetic (D) and non-diabetic (ND) obese subjects. Results are presented as median and interquartile range.

**Figure 4 ijms-24-17447-f004:**
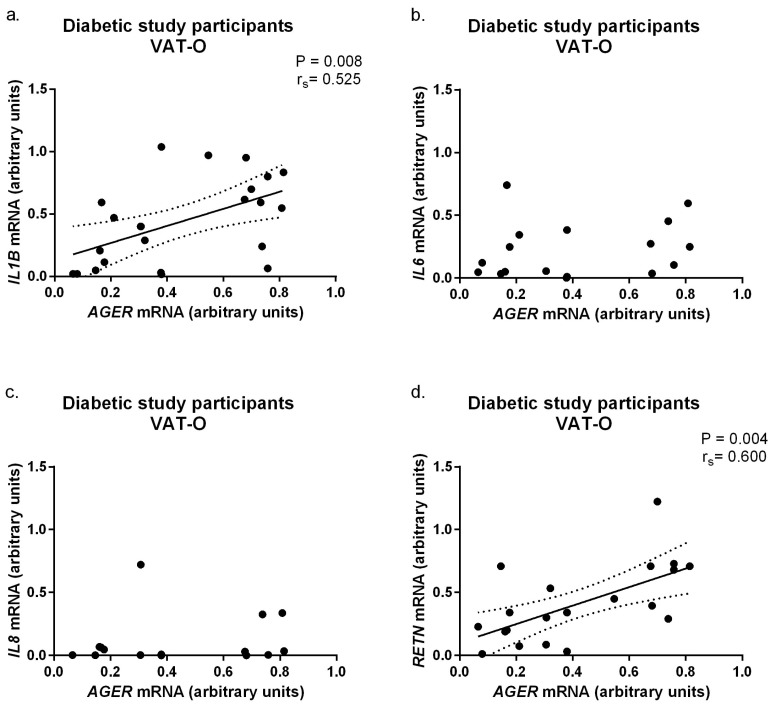
Correlations between AGER mRNA concentrations and mRNA levels of interleukin 1b (IL1B, (**a**)), interleukin 6 (IL6, (**b**)), interleukin 8 (IL8, (**c**)), and resistin (RETN, (**d**)) in the visceral adipose tissue of obese diabetic study participants (VAT-O). Black dots represent particular participants. Lines represent linear regression analysis with 95% CI interval.

**Table 1 ijms-24-17447-t001:** Clinical and biochemical characteristics of the study participants.

	Obese Individuals before Weight Loss (*N* = 62)		Obese Individuals after Weight Loss (*N* = 15)		Normal Weight Controls (*N* = 20)	
	Mean ± SD	Min–Max	Mean ± SD	Min–Max	Mean ± SD	Min–Max
Age (years)	40.48 ± 10.19	20–59	40.76 ± 8.13	29–59	44.08 ± 13.5	26–63
Weight (kg)	125.6 ± 17.11	87.8–170.0	73.11 ± 6.25	66.0–82.0	67.92 ± 10.48	54.0–74.0
BMI (kg/m^2^)	45.97 ± 5.70	35.43–63.42	28.2 ± 2.45	24.0–31.2	23.06 ± 1.33	20.7–24.93
Adipose tissue (% body mass)	48.54 ± 3.65	40.43–57.23	32.5 ± 3.4	26.0–37.0	-	-
Weight loss (kg)	-	-	32.4 ± 8.4	23.0–42.0	-	-
Glucose (mmol/l)	5.91 ± 1.37	3.67–9.85	4.82 ± 0.55	4.12–5.66	5.2 ± 0.3	4.8–5.4
HbA1c (%)	5.79 ± 0.51	5.06–7.20	5.1 ± 0.47	4.59–5.65	4.9 ± 0.28	4.6–5.3
Total cholesterol (mmol/l)	4.96 ± 0.91	3.13–7.87	4.76 ± 0.66	3.92–5.9	4.31 ± 0.20	4.24–4.78
LDL cholesterol (mmol/l)	3.04 ± 0.97	1.04–5.65	2.99 ± 0.35	2.54–3.58	2.82 ± 0.11	2.74–2.90
HDL cholesterol (mmol/l)	1.21 ± 0.25	0.59–1.78	1.38 ± 0.27	1.08–1.68	1.47 ± 0.26	1.24–1.76
Triglycerides (mmol/l)	1.43 ± 0.72	0.52–3.23	1.40 ± 0.38	1.02–1.85	1.32 ± 0.2	1.1–1.46
CRP (mg/l)	10.65 ± 5.29	1.21–21.61	4.51 ± 2.45	1.81–7.21	3.24 ± 1.72	0.7–5.20
Comorbidities	*N*	(%)	*N*	(%)	*N*	(%)
Hypertension	31	50.0%	5	33.3%	none	none
Type 2 diabetes mellitus/prediabetes *	23	37.0%	3	20.0%	none	none
Hyperlipidemia	34	55.0%	6	40.0%	none	none

BMI—body mass index calculated as weight (kg) divided by height squared (m^2^); CRP—C-reactive protein; HbA1c—glycated hemoglobin, HDL—high-density lipoproteins; LDL—low-density lipoproteins; N—number of subjects; * impaired fasting glucose and/or impaired glucose tolerance.

**Table 2 ijms-24-17447-t002:** Primers used for the analysis of lncRNA and gene expression at the mRNA level in adipose tissue.

Gene/lncRNA	Description		Primers	Ref.
*AGER*	advanced glycosylation end-product receptor	F	5′ TGTGCTGATCCTCCCTGAGA 3′	[51]
		R	5′ CGAGGAGGGGCCAACTGCA 3′	
lncAGER-1	long non-coding RNA AGER-1	F	5′ AACCAGGAGGAAGAGGAGGA 3′	[48]
		R	5′ TTGGCAAGG TGGGGTTATAC 3′	
*IL1B*	interleukin 1β	F	5′ CACCAAGCTTTTTTGCTGTGAGT3′	[19]
		R	5′ GCACGATGCACCTGTACGAT 3′	
*IL6*	interleukin 6	F	5′ CCTTCGGTCCAGTTGCCTTC 3′	[19]
		R	5′ GTGGGGCGGCTACATCTTTG 3′	
*IL8*	interleukin 8	F	5′ CACCGGAAGAACCATCTCACT 3′	[19]
		R	5′ TCAGCCCTCTTCAAAAACTTCTCC 3′	
*RETN*	resistin	F	5′ GCTGTTGGTGTCTAGCAAGAC 3′	[20]
		R	5′ CATCATCATCATCATCTCCAG 3′	
*ACTB*	β-actin	F	5′ CAGCCTGGATAGCAACGTAC 3′	[19,20,51]
		R	5′ TTCTACAATGAGCTGCGTGTG 3′	

F—forward, R—reverse.

## Data Availability

The data presented in this study are available on request from the corresponding author.

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
