# Peer review of "AGER-1 Long Non-Coding RNA Levels Correlate with the Expression of the Advanced Glycosylation End-Product Receptor, a Regulator of the Inflammatory Response in Visceral Adipose Tissue of Women with Obesity and Type 2 Diabetes Mellitus"

_ijms, 2023, doi:10.3390/ijms242417447_

Round 1
Reviewer 1 Report
Comments and Suggestions for Authors
The authors examined the expression levels of AGER and related factors in the adipose tissues of obese women. They observed elevated AGER mRNA levels in individuals with obesity or diabetes. Furthermore, a correlation was identified between lncAGER-1 and certain cytokine mRNA levels with AGER mRNA. One big limitation of the study is that the analyses primarily focused on RNA levels without assessing corresponding protein levels. Despite this, the study yields significant findings likely to capture the interest of both the scientific community and general readers.
The result of my search in Medline indicate that the topic is original, with no previously published papers related to the present subject. Analysis of clinically obtained samples, especially those collected before and after surgical operation, is challenging to acquire and adds significant value to the research results. This aspect is particularly noteworthy and holds the highest importance in this paper. Also, in my opinion conclusions included in this work are consistent with the evidence and arguments presented and they address the main question, which was posed.
One minor revision: the last sentence of the abstract doesn't make sense and requires modification.
Author Response
Reviewer 1
We would like to thank the reviewer for the positive reception of our work and the comment, which will undoubtedly improve the quality of the manuscript.
Comment 1
One minor revision: the last sentence of the abstract doesn't make sense and requires modification.
We thank the reviewer for this comment. In the revised version of the manuscript, we have changed the last sentence of the abstract, which we hope now makes the correct sense.
“In conclusion, obesity in women is associated with increased expression of AGER in SAT, while T2DM – with increased AGER and proinflammatory adipokines mRNA levels in VAT.” Lines 43-45.

Reviewer 2 Report
Comments and Suggestions for Authors
I carefully reviewed the paper titled "AGER-1 long non-coding RNA levels correlate with the expression 2 of AGER, a regulator of the inflammatory response in visceral adi-3 pose tissue of women with obesity and diabetes mellitus". Authors studied AGER-1 and lncAGER-1 in obese subjects. The paper has some scientific value but also has several issues that require revision.
I recommend authors using type 2 diabetes mellitus since it is more accurate terminology than type 2 diabetes.
Again in abstract, authors should make it clear "post-bariatric patients". Please be careful to make correct statements.
I recommend authors to avoid abbreviations in first mention. Spell out all in first use than give abbreviation in brackets, please.
Authors expressed the table 1 in methodology (...baseline clinical and biochemical parameters of the study participants are summarised in Table 1), however, I think this information belongs to the results section. Moreover, prefer "summarized" instead of "summarised".
Control group of 20 women is rather small study population. It can be stated as another limitation.
Author Response
Reviewer 2:
I carefully reviewed the paper titled "AGER-1 long non-coding RNA levels correlate with the expression 2 of AGER, a regulator of the inflammatory response in visceral adi-3 pose tissue of women with obesity and diabetes mellitus". Authors studied AGER-1 and lncAGER-1 in obese subjects. The paper has some scientific value but also has several issues that require revision.
We would like to thank the reviewer for the positive reception of our work and the comments, which will undoubtedly improve the quality of the manuscript.
Comment 1
I recommend authors using type 2 diabetes mellitus since it is more accurate terminology than type 2 diabetes.
We thank the reviewer for this valuable comment. In the revised version of the manuscript, we have replaced type 2 diabetes with type 2 diabetes mellitus in each case, including the manuscript title.
Title”AGER-1 long non-coding RNA levels correlate with the expression of AGER, a regulator of the inflammatory response in visceral adipose tissue of women with obesity and type 2 diabetes mellitus.”
“The advanced glycosylation end-product specific receptor (AGER) is involved in the development of metabolic inflammation and related complications in type 2 diabetes mellitus (T2DM).” Lines 28-29.
“Inflammatory cytokine and chemokine responses in a state of positive energy balance shift the immune system from an anti-inflammatory to a pro-inflammatory state and lead to an increased risk of developing metabolic disorders, including insulin resistance (IR) and type 2 diabetes mellitus (T2DM) [5].” Lines 59-63.
“On the other hand, its main strengths are its homogeneous, well-characterized population, with 62 participants in the study group, and its innovative nature, which can be considered an attempt to elucidate the mechanisms linking obesity and type 2 diabetes mellitus.” Lines 285-288.
“However, AGER-mediated development of metabolic inflammation, which predisposes to increased insulin resistance and type 2 diabetes mellitus diabetes, preferentially affects VAT.” Lines 338-340.
Subsequently, the abbreviation T2D has been replaced by T2DM, wherever it was necessary.
Comment 2
Again in abstract, authors should make it clear "post-bariatric patients". Please be careful to make correct statements.
We thank the reviewer again for pointing out this inappropriate expression. In the revised version of the manuscript, the following sentences have been modified:
“The expression of genes encoding AGER, selected adipokines, and lncAGER-1 was assessed by real-time PCR in visceral (VAT) and subcutaneous (SAT) adipose tissue obtained from 62 obese (BMI > 40 kg/m2, 24 diabetic) and 20 normal weight (BMI = 20-24.9 kg/m2) women and in 15 SAT samples from patients 18 to 24 months after bariatric surgery.” Lines 43-45.
“Due to the paucity of biological material, protein measurements were not performed in the group of patients after bariatric surgery” Lines 163-165.
In addition, the Figure 2 legend has been also modified.
Comment 3
I recommend authors to avoid abbreviations in first mention. Spell out all in first use than give abbreviation in brackets, please.
We would like to apologize for this evident mistake. In the revised version of the manuscript, all abbreviations have been spelled out in the first use.
Comment 4
Authors expressed the table 1 in methodology (...baseline clinical and biochemical parameters of the study participants are summarised in Table 1), however, I think this information belongs to the results section.
Following the Reviewer's valuable suggestion, Table 1 was moved to the Results section and the new sub-section entitled 2.1 Clinical and biochemical characteristics of study participants has been created. (Lines 93-115)
Comment 5
Moreover, prefer "summarized" instead of "summarised".
We apologize for this typo, it has been corrected in the revised version of the manuscript.
“The baseline clinical and biochemical parameters of the study participants are summarized in Table 1.” Lines 114-115.
Comment 6
Control group of 20 women is rather small study population. It can be stated as another limitation.
We do agree with the Reviewer that both – the low number of control patients and the low number of tissues obtained from the patients after bariatric surgery are chief limitations of our study that may hamper the value of the results. Subsequently, the following statement has been added to the part of the Discussion referring to the possible limitations of the study.
“Another limitation refers to the relatively small number of control tissues obtained from the normal weight individuals and patients who underwent bariatric surgery.” Lines 283-285.
